# Total Aseptization of Boar Semen, to Increase the Biosecurity of Reproduction in Swine

**DOI:** 10.3390/molecules26206183

**Published:** 2021-10-13

**Authors:** Ştefan Ciornei, Dan Drugociu, Liliana Margareta Ciornei, Mihai Mareş, Petru Roşca

**Affiliations:** 1Reproduction Clinic, Department Clinics, Faculty of Veterinary Medicine, Iasi University of Life Sciences (IULS), M. Sadoveanu Alee, no 6, 700489 Iaşi, Romania; stefan_ciornei@uaiasi.ro (Ş.C.); ddrugociu@uaiasi.ro (D.D.); l.ciornei@uaiasi.ro (L.M.C.); 2Laboratory of Microbiology, Department of Public Health, Faculty of Veterinary Medicine, Iasi University of Life Sciences (IULS), M. Sadoveanu Alee, no 6, 700489 Iaşi, Romania; mmares@uaiasi.ro

**Keywords:** bacteria, fungus, boar semen aseptization, progressivity, fluconazole, biosecurity of AI

## Abstract

The aim of the study was to establish the complete microbiological profile of boar semen (*Sus scrofa domesticus*) and to choose the most effective antiseptic measures in order to control and optimize AI reproduction in pig farms. One hundred and one semen samples were collected and analyzed from several pig farms. The microbiological profile of ejaculates was determined by evaluating the degree of contamination of fresh semen and after dilution with specific extenders. The bacterial and fungal load of fresh boar semen recorded an average value of 82.41/0.149 × 10^3^ CFU/mL, while after diluting the ejaculates the contamination value was 0.354/0.140 × 10^3^ CFU/mL. Twenty-four germs (15 bacterial and 9 fungal species) were isolated, the most common being *Candida parapsilosis/sake* (92%) and *Escherichia coli* (81.2%). Modification of the sperm collection protocol (HPBC) reduced contamination in raw sperm by 49.85% in bacteria (significant (*p* < 0.00001) and by 9.67% in fungi (non-significant (*p* < 0.111491). The load in bacteria and filamentous fungi can be controllable, but not in levuras fungi. Some fluconazole-added extenders (12.5 mg%), ensure fungal aseptization, and even an increase in sperm progressivity (8.39%) for at least a 12 h shelf life after dilution. Validation of sperm aseptization was done by maintaining sow fecundity unchanged after AI (insignificant *p* > 0.05).

## 1. Introduction

Studying the factors that may influence the biological value of semen subjected to preservation and artificial insemination (AI) is of great importance within the assisted-reproduction biotechnologies in pigs. Of these factors, an important role is attributed to the presence of microbial flora [1,2,3,4]. Changes of semen in terms of metabolism (glucose–sperm–germ relationship) may be caused by various germs types and their number per volume unit [1,5]. The presence of bacteria in extended semen is harmful for sperm viability and represents an infection risk for the inseminated sows [6,7]. Microorganisms have direct and indirect deleterious effects on sperm, altering their motility and fecundant capacity. Direct damage to sperm occurs by mechanical adhesion of colonies of bacteria or hyphae that lead to agglutination of sperm in doses, and the indirect effect refers to metabolic secretions secreted by germs, which can become toxic and kill the seminal cell. Fernandez et al. (2001), stated that there is a close correlation between sperm agglutination and level of semen contamination, and also that sperm agglutination might affect the fertility of inseminated sows. A diverse bacterial flora could decrease semen quality during preservation [8,9].

The most frequent bacteria types detected in boar semen are *E. Coli*, *Pseudomonas* spp., *Staphylococcus* spp., and *Proteus* spp. [5,8,10]. Some authors identified 56 strains—40 strains were detected in freshly collected and 16 in the extended semen. *Staphylococcus* spp. were identified in 75% of the raw semen samples, and 20% of the extended semen samples. *Streptococcus* spp. were identified in 60% and 10%, *E. coli* in 60% and 0%, *Pseudomonas* spp. in 30% and 25%, and *Micrococcus* spp. in 50% and 0%, respectively [1]. The presence of bacteria in stored semen demonstrates that current antibiotics are not fully efficient against them.

Several sources of contamination have been described, both of animal and non-animal origin. The first source of contamination is the boar (preputial fluid, skin, hair, and feces on the legs). The second source of contamination is the equipment used for semen collection and examination before processing [8,10,11]. Semen collection in farm animals is not a sterile procedure, with some bacterial flora contaminating the semen [6,8,10,12,13,14]. Dias et al. (2000), reported different germ loads depending on the hygiene and the method that was used for semen collection [12]. In case of a strictly hygienic collection, sperm contamination was between 490 and 975 CFU/mL, while in case of a normal collection, it was between 14.6 × 10^3^ and 18.8 × 10^3^ CFU/mL [12,13].

Most commercial extenders for boar semen contain the same groups of antibiotics, more exactly aminoglycosides such as gentamicin and lincosamides [1,14,15], to which some bacteria present in the semen become resistant [10]. Therefore, some bacteria survive the preservation process and are present in the insemination doses before AI.

To overcome the problem of semen contamination, some authors [3,8], recommend maintaining good hygiene and permanent application of a periodic disinfection protocol. In addition, it is necessary to use extenders that contain combinations of antibiotics that are effective against bacterial flora present in the semen and to conduct regular microbiological screening of boar semen in the swine industry to avoid the use of poor quality sperm [13]. Various products have been delivered in this regard, with acceptable efficacy. However, extenders with antifungal capacity have not yet been developed.

There is little information in the literature about fungal contamination of boar semen and its persistence during storage [16,17]. The usual conditions of storage before AI (liquid state at 17 °C for several days) represent excellent conditions for the development of fungi, and the extenders do not contain antifungal substances. Previous work has focused on bacteria, so little is known about the fungal contamination of boar semen and its effects on seminal parameters.

The identification of yeast and filamentous fungi in preserved semen confirms the idea that the antibiotics do not influence their presence and development [4]. Both in raw and extended semen, a number of fungi were identified. However, not enough studies regarding the degree of fungal contamination of boar semen have been performed, which justifies, once again, the appropriateness of this research.

This study aimed to highlight the bacterial and fungal contamination of boar semen during collection and the possibilities of complete aseptization of AI doses in sows, without negatively altering fertility.

## 2. Results

All the examined ejaculates were considered to be normal for each parameter of the spermogram. No appearance or pH inconformity was detected. The average volume of the ejaculates after filtration was 344.35 mL. Semen concentration varied between 0.18 × 10^9^ and 0.51 × 10^9^ spermatozoa/mL, with a mean of 0.36 × 10^9^ spermatozoa/mL; motility ranged between 65 and 86%, so the quality of semen was quite high. Other organoleptic parameters used in boar semen assessment were also normal taking in consideration the evaluation criteria described by Knox R.V. (2004) [18].

### 2.1. Microbiological Spermogram

The bacterial and fungal load of fresh boar semen recorded an average value of 82.41/0.149 × 10^3^ CFU/mL, while after diluting the ejaculates the contamination value was 0.354/0.140 × 10^3^ CFU/mL. The average microbiological profile in raw and diluted semen samples is presented in Table 1.

For raw semen, the number of bacteria and fungi varied between 22.4 × 10^3^ and 118.20 × 10^3^ CFU/mL, with a mean of 65.49 × 10^3^ CFU/mL in farm 1, while in farm 2, the average number registered a higher value 99.33 × 10^3^ CFU/mL, and the variation limits were between 74.60 × 10^3^ and 130.05 × 10^3^ CFU/mL. From all (101) semen samples, only 8 (7.9%) were undetectable contaminated (total viable aerobic count < 10^1^ CFU/mL). After dilution and 12 h storage at 17 °C, the average bacterial burden became 0.12 × 10^3^ CFU/mL (limits ranging between 0.08 × 10^3^ and 0.31 × 10^3^ CFU/mL) in farm 1, and 0.58 × 10^3^ CFU/mL (limits ranging between 0.14 × 10^3^ and 1.20 × 10^3^ CFU/mL) in farm 2, respectively. As it is shown in Table 1, the number of fungi per mL remained quite stable after the semen dilution, indicating a particular condition that must be evaluated. In farm 2, the degree of contamination was higher, apparently due to poor hygiene of sperm collection (boar-operator-room). Critical points of contamination were identified and described below in another experiment.

Concerning the qualitative aspects of semen contamination, the microbiological tests have highlighted a variety of bacterial and fungal species. Thus, in freshly collected semen, a total number of 14 bacterial and 9 fungal genera have been identified (Table 2). The species isolated with higher frequency were *Escherichia coli* (81.2% of samples) followed by *Staphylococcus* species (*aureus zooepidemicus*, *intermedius*, *hyichus*) (72.3%), *Pseudomonas aeruginosa* (63.4%), *Enterococcus* (*faecium*, *faecalis*) and *Streptococcus* (*suis*, other species) (45.5%), and *Proteus vulgaris* (35.6%). Other bacterial species such as *Tatumellaptyseos*, *Pantoea* spp., *Shigella* spp., *Yersinia* (*enterocolitica*, *rukeri*, *pseudotuberculosis*), and *Serratia* (*ficaria*, *marcescens*) were identified with a lower frequency (26.7%). *Bacillus* (*subtilis*, *cereus*, *megaterium*), *Arcanobacterium pyogenes* and *Actinomyces suis* were identified in 10.9% of the samples, and *Klebsiella pneumoniae* in 6.9%. 

Among fungal species, the yeasts belonging to *Candida* genus were the most prevalent in semen samples (*Candida parapsilosis* was isolated in 92.1% of samples). Other fungi were encountered in lower proportions: *Geotrichumcandidum* (72.3%), *Aspergillus* spp. and *Penicillium* spp. (63.3%), *Mucor racemosus* (45.5%), *Cladosporium cladosporioides* and *Fusarium* spp. (36.6%), and *Acremonium* spp. and *Alternaria alternata* (18.8%). Important to note is that after dilution and 12 h storage at 17 °C, yeasts such as *Candida parapsilosis* were identified in more 90% of AI doses.

### 2.2. Identification of Sources of Sperm Contamination

Before the introduction of a boar in the collecting room, the CFU (bacteria and fungi)/m^3^ of atmospheric air had lower values (30.9 × 10^3^ m^3^/air) compared to those obtained after collecting (44.2 × 10^3^ m^3^/air), (Table 3).

The flow of use of boars in the collection room in this farm is 5–7/day. Contamination of semen is dependent on the load of the external environment and the state of hygiene of the genital tract. The most common sources of contamination of semen are the foreskin, the atmospheric air in the collecting room, and the laboratory equipment used [19].

Due to spraying in the atmosphere of the room, dust and microorganisms settle quickly and those present on the skin and limbs of the boar will adhere to it and will not rise into the atmosphere. The microbiological load of the atmospheric air in the collecting room will be reduced in this way.

After the application of HPBC protocol it was observed that after obtaining the seminal material and its dilution both the number of bacteria and of fungi were much lower (Table 4). Thus the bacteriological load of the raw sperm decreased by 49.85% (from 62.39 × 10^3^ to 31.29 × 10^3^ CFU/mL), and in that after dilution it decreased by 87.2% (from 0.125 × 10^3^ to 0.016 × 10^3^ CFU/mL). But the fungal load followed a smaller reduction of 9.67% (from 0.115 × 10^3^ to 0.140 × 10^3^ CFU/mL) in raw semen and 9.15% (from 0.153 × 10^3^ to 0.139 × 10^3^ CFU mL) after dilution and these were represented by yeast fungi.

### 2.3. Antifungal Aseptization of Sperm by Potentiated Extenders

The effect of different concentrations of fluconazole on sperm extenders (A and B) are given by calculating the average of the most important types of sperm motility in Table 5. The mean values of the control samples (M) for extender A were 80.3% in motility (M%) (with limits between 72% and 82%), and the progressivity (P%) was 39.3%, (with limits between 36% and 42%). In the case of extender B, the mean was 75.6% (with limits between 47% and 77%), for total motility, and 30.6% (with limits of 30% and 32%) for progressivity. Concentration of 25mg% fluconazole (E1) on extender A produced an M% of 72.0% and a P% of 39.0%, and at extender B, an M of 64.3% and a P of 20.3%. A concentration of 12.5% fluconazole (E2) on extender A produced an M% of 80.6% and a P% of 42.6%, and at extender B, an M of 73.0% and a P of 29.3%.

In the group with 25 mg% fluconazole (E1) for extender A the average value of motility (M%) (72.0%) was 11.5% lower than the average of the control group (80.3%), and for progressivity P% it was 0.7% lower (39% vs. 39.3%), for extender B the average value of M% was 29.1% lower (64.3% vs. 75.6%), and for P% with −50.7% (20.3% vs. 30.6%).

In the group with 12.5 mg% fluconazole (E2) for extender A the average value of M% compared to the control group was 0.37% higher (80.3% vs. 80.6%), and for P%, 39% higher (39.3% vs. 42.6%); for extender B the average value of M% was 3.5% lower (75.6% vs 73%), and for P% it was 4.4% lower (30.6% vs. 29.3%).

For the group E1 and the extender A, the motility was significantly lower than in control group (*p* = 0.0137) while the progressivity did not differ significantly (*p* = 0.265).

For the group E1 and extender B, both motility and progressivity differed significantly from the control group (*p* = 0.00632 and *p* = 0.00018, respectively).

For the group E2, the difference was statistically significant only for extender A and progressivity (*p* = 0.0297).

### 2.4. Testing Semen Extenders Added with Fluconazole by Determining Sperm Progressivity in Storage Dynamics

To observe the effect of the antifungal substances on the quality of sperm (mobility indices) during the preservation of insemination doses, we followed the total motility and progression in dynamics for 48 h (Table 6).

At 12 h after dilution, the motility (M%) of the samples from the control group (M) had values between 82% and 78% with an average of 80%. Of the sperm with total motility, an average of 39% of them had forward movements (progressivity); the limits being between 42% and 36%. Preservation led to a steady but sustained decrease in M% and P%.

Within the group E1 (25 mg% fluconazole) an average value of 72.5% was obtained for M%, lower than M (80%); instead, the average percentage of P% registered equal values (39%). The variation limits were between 47% and 31%.

The mobility parameters studied in dynamics registered progressive decreases as the conservation duration increased in both control and experimental groups. Thus, the determinations performed at 24 h after dilution, showed average values of motility (M%) in the experimental groups (E1 and E2), lower (67% and 65%, respectively), than those recorded in the control (69.5%). P% was (27% and 30%) vs. 31.5%.

The results of the determination performed at 48 h show lower average values of progressivity (P%) at E1 and E2 (19.5% and 16%), compared to the mean value at control (25%). The decrease of the average values of M% and P%, registered at both E.Ls. (experimental lots), both at 24 and 48 h after dilution, correlated with the decrease of the values of these parameters also at M (control lot), a decrease which was achieved in a smaller proportion.

### 2.5. Validation of Additive Extenders by AI on the Farm

For a better appreciation of our research, the effect of the new extender formulas on the reproduction indices was also followed. The aim was to observe the effects of the antifungal in the extender (12.5% fluconazole) on the fecundity and prolificacy of the sows; the results are presented in Table 7. The prolificacy of the lots was slightly different, with a higher value in the control group. The average number of piglets obtained at farrowing was 9.125 in C.G., 0.43 less than the average per farrowing in E.G. (9.062). For the control group (C.G.) the total number of piglets obtained from all sows was 146, with one piglet less than those obtained in the experimental group (E.G.) (145).

## 3. Discussion

The average bacterial burden determined for all farms in our study (82.4 × 10^3^ CFU/mL in raw and 0.354 × 10^3^ CFU/mL in diluted semen) was similar to that reported by others [5,20], and depended on the hygiene of semen collection and handling process and also on antibiotic resistance of the microbiota from a particular farm. Values were lower than those obtain by manual collection with double glove (384 × 10^3^ CFU/mL) or by automatic collection (349 × 10^3^ CFU/mL) [21]. Similar results have been published for other species. For example, Wiersbonski (1981) found that the bacterial load of bull semen used for A.I., varied between 1.5 × 10^6^ and 6.5 × 10^6^ CFU/mL [22]. Rota et al. (2011) emphasized the presence of bacteria and fungi in stallion semen and other reproductive segments (presence and distribution) [23].

High bacterial count in AI doses has been associated with a decline of sperm quality [24], reduction of fertility [4], and transmission of pathogens into the females [25]. Therefore, current legal guidelines require the addition of antibiotics to each AI dose (Council Directive, European Union, 90/429/EEC). However, the added antibiotics are often no longer sufficient to stop bacterial growth in AI doses as antimicrobial resistance becomes more widespread [19,26]. Lately, many efforts in the search for promising possibilities to replace antibiotics in liquid-preserved boar semen have been made to reduce the development of multi-resistant bacteria by minimizing the selection pressure of antibiotics. Approaches include, but are not limited to: colloid centrifugation [27], supplementation of antimicrobial peptides [26,28,29,30], addition of algal extracts [31], low-temperature preservation [22,29,32], and development of a new antimicrobial concept added to the extenders [33].

In both diluted and raw semen there were identified different kinds of fungi. The literature concerning the fungal contamination of boar semen is relatively poor and it could be a justification for the opportunity of our study. Concerning the origin of fungal contaminants, we could identify two sources—the indoor air and the boar’s prepuce. Generally, mold species such as *Mucor*, *Aspergillus*, *Penicillium*, *Cladosporium*, *Acremonium*, *Fusarium*, and *Alternaria* are airborne fungi, while yeasts such as *Candida parapsilosis* are skin commensals. The airborne fungi may occur in semen in different concentrations dependent on their indoor level and we can reduce them by rigorous hygiene during collection and in processing rooms. The yeasts’ concentration in semen samples was quite consistent indicating an internal source of contamination.

Semen contamination was detectable in more than 90% of samples, with only about 8% of samples exhibiting values less than 10 CFU/mL. Some authors reported similar or higher percentages in their papers [4,9,34]. Dias et al. (2000) reported different loads of germs depending on hygiene and collection method. Using a collection process with strict hygiene, semen contamination was between 490 and 975 CFU/mL, while using a current collection procedure it ranged between 14.6 × 10^3^ and 18.8 × 10^3^ CFU/mL [12].

Most commercial extenders designed for boar semen dilution contain antibiotics belonging to the same group (aminoglycosides such as gentamicin and lincomycin) [2,14,15]; some bacteria present in semen could be resistant to these antibiotics [10,34]. In our study, the decrease of bacterial burden was extremely efficient after dilution (more than 2 × log_10_), indicating a strong antibacterial effect of the extenders. On the contrary, even the ratio used for fresh semen and extender was approximately 1:7, the fungal burden did not decrease after dilution and 12 h storage. Moreover, yeasts such as *Candida parapsilosis* multiplied during the storage period due to a favorable chemical composition of the semen extenders (all of them contain glucose and other sugars). This fact is strongly indicative of the lack of antifungal activity of the extenders that contain only antibiotics and no fungistatic or fungicide substances. 

Some genera of bacteria and fungi were identified in the air in the room, but also in the prepuce and in the freshly harvested semen. We assume that the genera identified in the seminal material, being present in the other samples analyzed, came from the external environment. Contamination occurs with the contact of the sperm jet with the air and the collection equipment. This is due to the peculiarities of ejaculation in the boar, with reference to time and volume of semen.

The increase in the number of microorganisms is due to the movements of boars in the collection room. Along with the agitation of males caused by libido and sexual reflexes, microorganisms are trained in the air which subsequently sediment and produce contamination. Microorganisms and dust accumulate and float in the atmosphere of the room. Due to the peculiarities of ejaculation in the boar and of the collection equipment (filter container), the sperm jet enriched with such particles when passing through the air, also entails those already sedimented on the filter, thus producing its contamination.

Establishing the most important places where sperm can be microbially contaminated creates real possibilities for limiting the contamination of semen with germs. By applying the HPBC boar semen collection protocol, it confirms its success by reducing sperm contamination from both external and internal sources.

The yeasts, being determined constantly and relatively in the same number, we consider that they have an internal origin of sperm contamination, instead, the filamentous fungi that come from external sources of contamination, were missing, which makes us say that by following the collection protocol, completed by us and supplemented by the observance of hygiene conditions, it is possible to reduce the sperm CFU at critical points of contamination.

After identifying and removing the critical points of sperm contamination during the collection process, we can appreciate that the microorganisms left in the sperm have endogenous origin being represented in their vast majority by yeasts. For more effective aseptization, an antifungal preparation (fluconazole) was introduced into the formula of the extenders used.

The computerized sperm analysis system (CASA) provides information about the mobility of each sperm cell, by processing electronic images of sperm, it reconstructs the trajectory of each sperm cell, and simultaneously and objectively evaluates each component of the sperm so that even minor changes in their mobility can be detected. Depending on the type of chambers used (Leja, Mofa) there may be differences in reporting [35,36,37,38]. The CASA method has proven to be more accurate and correct, and therefore more objective, than the classical evaluation [39]. Ibănescu (2016) described that the type of analysis chamber used in the CASA method may have different effects on the values obtained [38].

The differences between control group M and experimental groups E1 and E2 are calculated and presented as percentages in Table 5.

It appears that the concentration of 25 mg fluconazole in both extenders (A and B) had a probable toxic effect on sperm leading to decreased indices of mobility M% (*p* = 0.0137/*p* = 0.00632) and P% (*p* = 0.265/*p* = 0.00018). Different concentrations of fluconazole were tested by probing, indicated by antifungal results. At a higher concentration (25 mg%) all sperm mobility determinations had the lowest values for both the two extenders, but also in the conservation dynamics. Because the lower concentrations were close to the control, we consider that a concentration around 25 mg% leads to the biochemical change of the sperm environment, which ultimately reduces their total mobility.

In contrast, the concentration of 12.5 mg% fluconazole in extenders had a totally different effect; in extender B there was a small decrease in both P% and M% and in extender A there was a probable metabolic activation of sperm P%, which increased more than M%. For the group E2 (12.5 mg% fluconazole), the difference was statistically significant only for extender A and progressivity (*p* = 0.0297).

Fluconazole at a concentration of 12.5 mg% in extender A exerted a stimulating role on sperm at least for a storage period of 12 h, after dilution, at a temperature of 17 °C.

Following the artificial insemination of the sows in the respective groups, four sows, both from the C.G. as well as from the E.G., manifested the return to heat syndrome (Table 7). Thus, the fecundity of the experimental group was 80%, a value identical to that of the control group (80%), so we can say that there were no visible changes in the processes of fertilization and nesting. There were no statistical differences (*p* > 0.05) between the fecundity and prolificity occurring in the control and experimental groups. Moreover, the sows nursed and weaned the piglets under normal conditions, and then resumed their reproductive activity on the farm

Thus, we can say that changing the formula of extenders by adding fluconazole at a concentration of 12.5 mg/L, brings a real benefit especially in the case of sperm heavily contaminated with yeast from the prepuce during harvest, by offering better possibilities for aseptic handling of semen intended for preservation, and the action of the antifungals does not adversely affect the quality of sperm or reproductive evidence. 

The technique of collecting boar semen is well known; a decisive role in the contamination of the sperm is represented by the hygiene of the process and the observance of the respective stages. However, the microbiological load in the sperm persists even after dilution of the semen. Regarding the qualitative microbiological spermogram of the diluted seminal material destined for preservation and insemination, an appreciable proportion has yeasts compared to bacteria, an aspect that also corresponds to the quality of the extender, showing the possibility of a total aseptic extender.

For better aseptization of the seminal material without significantly altering the longevity and fertilizing capacity of the sperm, we propose and recommend the introduction, in the extender formula in addition to antibiotics, an antifungal preparation.

The lack, in the domestic and international literature, of data on contamination and the possibility of fungal aseptization of boar semen during conservation, led to the motivation to try to modify or supplement the formula of extenders used to preserve boar semen.

The purpose of the study was to find a way to reduce and eliminate contamination of boar semen. Through the developed hygiene protocol and the addition of extenders, we managed to control the biosecurity of a boar semen. By the fact that the reproduction indices of the sows (fecundity and prolificity) were similar, the total aseptization was validated and we obtained a safe semen, destined for the conservation and sowing of the sows.

## 4. Materials and Methods

All procedures involving animals were carried out in accordance with guidelines and regulations according to the European Commission Directive for Pig Welfare, and IULS and farmers.

### 4.1. Pig Farms

The studies were carried out in three pig farms in Romania distributed geographically as follows: Farm 1 in the N.E., Farm 2 in the S.E., and Farm 3 in the S.W. All research protocols were staged and carried out in succession over several years, but before the outbreak and evolution of African Swine Fever in Europe. Farms are commercial complexes with a large flow of animals and any influence on breeding rates can positively/negatively influence the farm’s economy. Breeding management in pig farms plays an important role in producing the number of piglets for fattening and slaughter. 

### 4.2. Biological Material

In all farms, the genetic material used for breeding was represented by boars and sows of high genetic value, which cross-produce and deliver line piglets from fatteners. All conditions of specific biosecurity, deworming, nutrition, and well-being were met. The maintenance and exploitation of the animals was optimal, in accordance with European standards. At the time of each experiment, the animals were in good clinical health.

### 4.3. Reproduction Organization

Within the farms there is a reproductive laboratory where andrological and gynecological activity is planned. After a well-established daily schedule, sows are identified in the heat, the required number of doses for AI/AI repetition is calculated, a sufficient number of boars are distributed for collecting, ejaculates are collected, and doses for AI and storage are produced. The technological process of collection, examination, dilution, preservation, and insemination of semen were generally similar.

#### 4.3.1. Boar Semen Collection

The main method of sperm collection was used—the manual method with collector cup and filter. The protocol took place in a specially designed space—the collection room—equipped with a mannequin. The success of the harvest and the quality of the ejaculates varied between farms, depending on the ability and training of the operators. Before collecting the sperm, the operators have the obligation to groom the prepuce and to empty the preputial diverticulum of the boars.

#### 4.3.2. Examination of Ejaculate Quality

All samples were examined according to the standard methodology [18] and grouped into two categories: general examinations (organoleptic and macroscopic) and special examinations (microscopic). Appearance was evaluated in a transparent glass, examining the degree of turbidity and possible presence of the blood or other unusual colors. The smell was evaluated directly over the recipient of semen, searching for the usual smell, the presence of urine, or other unexpected odors. Concentration was assessed using the AccuRead Sperm Counter (IMV-Technologies, L’Aigle, France) [11,40]. Randomly, and when the results of the classical spermogram indicated differences., the quality of the ejaculates was also evaluated by the CASA system (computer assisted sperm analysis) [38].

#### 4.3.3. Dilution of Semen

Compliant ejaculates were diluted with commercial extenders. Extender A ensures long-term sperm viability (5–7 days) and contain an association of antibiotics; the manufacturer does not disclose which antibiotics are included. Extender B is a medium extender (3–5 days) and contains only one antibiotic, gentamicin. Extenders differ depending on the composition and shelf life at 17 °C. The semen extenders were chosen as the best/used on the market at that time [18,40].

#### 4.3.4. Preparation of AI Doses and Their Preservation

After the classical examinations and the determination of the volume and concentration, the dilution ratio was calculated/indicated automatically. For each insemination dose, 3.5 billion (3.5 × 10^9^) sperm were assigned in a volume of 80 mL. Those that were to be used to repeat AI were stored in incubators at a temperature of 17 °C for up to 5 to 7 days, depending on the type of sperm extender [13].

#### 4.3.5. Artificial Insemination

In this phased study, AI was performed in sows to monitor reproductive parameters, such as fecundity and prolificacy. The insemination method was classical, with fixation of the endocervical catheter (EC AI), according to the technique described by Hafez E. (2000) and then by Knox R. (2004) [18,40].

### 4.4. Microbiological Spermogram

The research was performed on 20 boars used for artificial insemination from Farm 1 and Farm 2. For the microbiological examination the samples were taken from freshly collected semen (raw) and semen after dilution and its distribution in doses for preservation (over 100 tests). The samples were collected in sterile vials and individualized (for identification) with the serial number on the boar’s ear tag. The determinations were made on freshly collected semen and after dilution and 12 h storage at 17 °C.

#### 4.4.1. Quantitative Determinations (CFU)

The semen samples (3 mL) were collected aseptically from each ejaculate before and after dilution, and transferred into a sterile container until the start of the test. The total viable aerobic count, also known as total number of viable bacteria and fungi was assessed using the serial dilution method and incubation in aerobic conditions. A series of ten-fold dilutions (10^−1^, 10^−2^, 10^−3^) of the semen samples was performed using tubes containing phosphate buffer saline (Biokar, Allonne, Oise, France). From each dilution, six volumes of 100 μL were plated on six petri dishes containing solid media. Three of them contained tryptone soy agar (Biokar, France) and were used for the enumeration of bacteria after an incubation of 24 h at 37 °C and the others containing Sabouraud Chloramphenicol Agar (Biokar, France) were used for the enumeration of fungi after an incubation of 3 days at 25 °C. Finally, the bacterial and fungal burden in raw and diluted semen was calculated and expressed as CFU/mL, according to methods by APHA [41,42]. The total viable aerobic count per mL consisted of CFU number of bacteria and CFU number of fungi.

#### 4.4.2. Qualitative Determinations (Typification of Bacterial and Fungal Genera)

In order to differentiate the bacterial species contaminating both the raw and diluted semen, each sample was plated onto Columbia agar with 5% (*v*/*v*) sheep blood and Mac-Conkey agar, respectively. The plates were incubated in aerobic conditions at 37 °C for 24 h. After this period, one colony of each morphotype was transferred onto tryptone soy agar and re-incubated for 24 h at 37 °C in order to obtain a fresh culture ready for identification. The identification of bacteria isolates was performed using Gram stain and ID32E, ID32GN, ID32STAPH, and ID32STREP (bioMérieux, Craponne, France). Filamentous fungi also called molds were identified on the basis of macroscopic and microscopic features using the primary cultures onto Sabouraud Chloramphenicol Agar. The yeasts were identified by biochemical tests using ID32C strips (bioMérieux, France) [41,42].

### 4.5. Identifying the Sources of Sperm Contamination and Eliminating Them

#### 4.5.1. Critical Points of Sperm Contamination

In order to identify the critical points of contamination, an experiment was carried out in Farm 2 where the possible sources of contamination (internal and external) were followed. The samples were taken from the freshly collected semen, from the mucosa of the external genitalia, from the prepuce wash, from the apparatus, and from the atmosphere of the collection room. The Koch sedimentation method was used to determine the number of bacteria in the air of the room/m^3^, and Omelianski’s formula was used for fungi according to methods proposed by APHA [41,42,43].

#### 4.5.2. Possibilities to Optimize the Decrease of Contamination in Critical Points by HPBC Protocol

The hygiene protocol and the biosecurity of sperm collection (HPBC) was developed and implemented. As a precursor to the boar semen collection techniques, a mist spray-fog of lightly decontaminating substances (Misoseptol) was introduced 15 min before the start of the collection. Misoseptol, a decontaminant based on essential oils, was used to reduce the microbial burden in air before to start the semen collection. The operator equipped with two latex gloves cleaned the toilet and sanitized the prepuce region, obligatorily emptying the prepuce diverticulum of secretions and urine. Then the first glove was removed and collection was started with the remaining glove. The sanitation and control tests were resumed before and after the application of this strict hygiene protocol of the sperm collection process, although they made a number of 80 determinations.

### 4.6. Antifungal Aseptization of Sperm by Potentiated Extenders

For more effective aseptization, the introduction of an antifungal drug (fluconazole) into the commonly used formula extenders was investigated. The antifungal introduced was chosen according to the antifungal and water solubility results, and its minimum concentration was given by the minimum inhibitory dose (0.5 mg/L) obtained by ATB-Fungus3 antifungals (Bio Merieux, France).

The research was carried out on freshly collected semen, from ten boars used for artificial insemination from farm 2. Two extenders, A (long preservation time) and B (medium) were used, which were added with different concentrations of Flu (Fluconazole): Lot E1 25 mg%, Lot E2 12.5 mg% and compared with control lots (without Flu). The assessment of sperm compatibility was determined by assessing motility 12 h after dilution by CASA, (Hamilton Thorne, Beverly, MA, USA HT-USA), with the methodology of working on a Leja chamber [37,44]

### 4.7. Testing Semen Extenders Added with Fluconazole by Determining Sperm Progressivity in Storage Dynamics

The experimental protocol aimed at evaluating some sperm parameters in the dynamics of semen preservation, of semen extenders with fluconazole. Ejaculates of two boars from farm 1 were used, after resuming the methodology of the previous experiment (M, E1 Flu.-25 mg%, E2 Flu.-12.5 mg% in extenders A and B). Doses of semen as per AI (80 mL) were stored in a thermostat (17 °C) for 3 days (average storage of doses on farms). At 12, 48, and 72 h the motility (M% and P%) was evaluated with the CASA system (Telos software version 14, IMV France) and the methodology of the Mofa chamber [35,36,38].

### 4.8. Validation of Additive Extenders by AI on the Farm

The research was carried out on farm 3. The experimental group (L.E.) was 20 sows, inseminated with 3.5 × 10^9^ sperm/80 mL extender with 12.5 mg% fluconazole, and the control (LM) 20 sows inseminated with 3.5 × 10^9^ sperm/80 mL simple extender. The groups were made up of sows that spontaneously manifested their estrous phase, and were artificially inseminated by the classical method. The conditions of maintenance, feeding, hygiene, and comfort were identical for the respective lots. The gestational diagnosis was established by ultrasound 21 days after the date of sowing. The device used was a portable Agroscan with a fixed probe of 3.5 and 5 MHz.

### 4.9. Statistical Analyses

Basic descriptive statistics, chi-square test, and one-way ANOVA test were used to interpret the obtained data. For the ANOVA test, *p* < 0.05 was considered statistically significant. Statistical analysis was done with Prism version 8 (GraphPad Software 5.0., La Jolla, CA, USA, www.graphpad.com, accessed on 2 October 2021)

## 5. Conclusions

Boar semen collected in usual conditions contains a series of aerobic germs, bacteria (62.5%) and fungi (37.5%). The antibiotics used in commercial extenders and a rigorous hygiene during semen collection and processing can significantly reduce the bacterial contamination of A.I. doses. However, the fungal contamination persists in diluted semen stored at 17 °C for 3–7 days; the yeasts representing these microorganisms were constantly isolated in such samples. The yeasts cannot be removed from the semen by either addition of antibiotics or by optimizing the collection process hygiene. Fluconazole-added extenders (12.5 mg%), ensure the solution of this problem, and there is even a stimulating role on sperm with an increase in sperm progressivity (8.39%) for at least a 12 h shelf life after dilution.

Similar to G. Althouse who described and introduced the term “bacteriospermia”, in referring to isolated and identified bacteria in fresh and diluted semen in our study, the term “mycospermia” was introduced, referring to the fungal contamination of boar semen.

## Figures and Tables

**Table 1 molecules-26-06183-t001:** Microbiological profile in fresh and diluted semen in a two swine farms in Romania.

The Origin of Semen Samples		Bacterial Burden (10^3^ CFU/mL)		Fungal Burden (10^3^ CFU/mL)	
		Raw Semen	Diluted Semen	Raw Semen	Diluted Semen
Farm 1	Extender A *	65.49	0.120	0.103	0.089
Farm 2	Extender B **	99.33	0.588	0.194	0.192

* Extender A: long-term preservation, contains a cocktail of antibiotics; ** Extender B: medium-term preservation, contains only one antibiotic.

**Table 2 molecules-26-06183-t002:** Frequency of bacterial and fungal species isolated from boar semen.

	Genus	Species	Frequency of Isolations % (n)
**Bacteria**	*Escherichia*	*coli*	81.2 (82)
	*Staphylococcus*	*aureus, zooepidemicus, intermedius, hiyicus*	72.3 (73)
	*Pseudomonas*	*aeruginosa*	63.4 (64)
	*Streptococcus*	*suis*	45.5 (46)
	*Enterococcus*	*faecium, faecalis*	45.5 (46)
	*Proteus*	*vulgaris*	35.6 (37)
	*Yersinia*	*enterocolitica, ruckeri, pseudotuberculosis*	26.7 (27)
	*Tatumella*	*ptyseos*	26.7 (27)
	*Pantoea*	spp.	26.7 (27)
	*Serratia*	*ficaria, marcescens*	26.7 (27)
	*Shiqella*	spp.	26.7 (27)
	*Actinomyces*	*suis*	10.9 (11)
	*Bacillus*	*subtilis, cereus, megaterium*	10.9 (11)
	*Arcanobacterium*	*pyogenes*	10.9 (11)
	*Klebsialla*	*pneumoniae*	6.9 (7)
**Fungus**	*Cladosporium*	*cladosporoides*	36.6 (37)
	*Penicillium*	spp.	63.3 (64)
	*Fusarium*	spp.	36.6 (37)
	*Aspergillus*	spp.	63.3 (64)
	*Mucor*	*racemosus*	45.5 (46)
	*Alternaria*	*alternata*	18.8 (19)
	*Geotrichum*	*candidum*	72.3 (73)
	*Acremoniu*	spp.	18.8 (19)
	*Candida*	*parapsilosis, sake*	92.1 (93)

**Table 3 molecules-26-06183-t003:** Microbiological load in the air collection room before and after the procedure for collecting sperm from boars.

m^3^/air	Before Collection		After Collection	
		Total		Total
Bacteria	19.7 × 10^3^	**30.9 × 10^3^**	26.8 × 10^3^	**44.2 × 10^3^**
Fungus	11.2 × 10^3^		17.3 × 10^−2^	

**Table 4 molecules-26-06183-t004:** Microbiological spermogram before and after the use of hygiene protocol and the biosecurity of sperm collection (HPBC).

	Bacterial Burden (10^3^ CFU/mL)		Fungal Burden (10^3^ CFU/mL)	
	Raw Semen	Diluted Semen	Raw Semen	Diluted Semen
**Before (HPBC)**	62.39	0.125	0.155	0.153
**After (HPBC)**	31.29	0.016	0.140	0.139
**CFU Evolution (%)**	**−49.85%**	**−87.2%**	**−9.67%**	**−915%**
**Statistical Significance (*p*)**	**<0.00001 (yes)**	**0.000726 (yes)**	**0.111491 (no)**	**0.08078 (no)**

HPBC = Hygiene Protocol and the Biosecurity of sperm Collection. HPBC is highly effective against bacterial contamination of semen—both raw (*p* < 0.00001) and diluted (*p* = 0.000726)—but seems to have no effect against fungal burden (*p* > 0.05).

**Table 5 molecules-26-06183-t005:** Assessment of compatibility between semen extenders and different concentrations in fluconazole.

	Extender A		Extender B	
	M%	P%	M%	P%
**M**	80.3	39.3	75.6	30.6
**E1**	72.0	39.0	64.3	20.3
**E2**	80.6	42.6	73.0	29.3
**(%) M vs. E1**	−11.5% *p* = 0.0137	−0.7% *p* = 0.265	−29.1% *p* = 0.00632	−50.2% *p* = 0.00018
**(%) M vs. E2**	+0.37%	+8.39% *p* = 0.0297	−3.5%	−4.4%

M%, total sperm motility; P%, total sperm progressivity; E1, fluconazole 25 mg%; E2, fluconazole 12.5 mg%; M, martor/control. Extender A: long-term preservation 5–7 days, Extender B: medium-term preservation 3–5 days. *p* < 0.05 was considered statistically significant.

**Table 6 molecules-26-06183-t006:** Testing semen extenders added with fluconazole by determining sperm progressivity in storage dynamics.

Preservation Time		12 h		24 h		48 h	
		M%	P%	M%	P%	M%	P%
**V1**	**M**	82	42	70	27	47	14
	**E1**	82	47	73	33	46	17
	**E2**	84	46	65	27	42	12
**V2**	**M**	78	36	69	36	70	36
	**E1**	63	31	61	21	51	22
	**E2**	78	39	65	33	46	20
**X**	**M**	80	39	69.5	31.5	58.2	25
	**E1**	72.5	39	67	27	48.5	19.5
	**E2**	81	42.5	65	30	44	16

M, control lot; E1, experimental lot 1 (fluconazole 25 mg%); E2, experimental lot 2 (fluconazole 12.5%); M%, total motility; P%, progressivity (fertilizing capacity). Semen preservation time at +17 °C, 12, 24, and 48 h; V1 and V2, boar ejaculates; x, average.

**Table 7 molecules-26-06183-t007:** Fertility and prolificity of AI sows with fluconazole additive extenders.

Group	Number of Gilts AI (n)	Sperm/Dose × 10^9^	Fecundity		Prolificity	
			Nr.	%	Total Piglets (*n*)	Average/ Farrowing
**C.G.**	20	3.5	16	80	146	9.125
**E.G.**	20	3.5	16	80	145	9.062
**Statistical significance *p* = 0.279598**						

C.G., control group; E.G., experimental group. There were no statistical differences (*p* > 0.05) between the fecundity and prolificity occurring in the control and experimental groups.

## Data Availability

The data obtained are available at the IULS university/Clinics/Reproduction Department in Iasi, Romania.

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
