# Peer review of "Total Aseptization of Boar Semen, to Increase the Biosecurity of Reproduction in Swine"

_molecules, 2021, doi:10.3390/molecules26206183_

Round 1
Reviewer 1 Report
In this study, Ciornei and colleagues analyzed the microbiological profile of boar semen, studying new strategies to reduce semen contamination from bacteria and fungi. They found that the bacterial and fungal load of fresh boar semen, collected in different Romanian farms, recorded an average value of 82.41/0.149x103CFU/mL, while after diluting the ejaculates with extenders the contamination value was 0.354/0.140 x103CFU/mL. Adopting some techniques to modify the sperm collection protocol (HPBC) reduced contamination in raw sperm (by 49.85% in bacteria and by 9.67% in fungi). To reduce fungal contamination, they added Fluconazole to extenders, which also increased sperm progressivity.
Although the paper is interesting, it needs an extensive revision prior its publication in Molecules:
Abstract
- It is not properly clear which animal they used for they study (pig and boar are both indicated); I suggest to add the scientific name of the species;
- In line 12, it should be better to indicate the exact number of the collected samples;
- In line 13, “determined by determining” should be changed;
- In line 16, the exact number of bacterial and fungal species should be given.
Introduction
- The cited literature in brackets should start from 1;
- In line 29, please specify the meaning of “in term of metabolism”;
- In line 32, add more details concerning these “direct and indirect” deleterious effects on sperm;
- In the last part, the aim of the study should be better explained
Results
- In Table 1, it should be indicated if the data refers to extender A or B (and, in case, if there were differences between them),
- In Table 2, it should be interesting if the authors could add also a quantitative data (how many coli etc);
- The Table 4 is confusing, since it is not clear the choice of the used criteria;
- In Table 7, it should be easier for the reader if, instead of T1, T2 and T3, the exact time would be indicated;
- In the results, the statistical analysis is completely missing (but the authors cite it in line 182);
Discussion
- Lines 200-206 are results, so they should be put elsewhere, moreover, lines 293-300 are repetition of the results and they can be avoided in this section;
- In line 289, the year of the cited study should be added;
- The authors should better explain why, in their opinion, 25 mg of Fluconazole induce a decrease in sperm motility (line 302);
- Since the authors did not find difference in sperm quality or reproductive evidence (line 317), they should provide more explanation on the aim and significance of their study;
Materials and Methods
- In line 338, please explain the meaning of S-V (I suppose it should be S-W, south-west);
- In line 371, the meaning of “when needed” should be clarified;
- The sentence in line 432 seems truncated;
- It is not clear why the authors did not use all the samples for all the experiments (i.e. they used just the samples from farm 1for Fluconazole experiment, and just animals from farm 3 for AI).
Conclusion
The sentence in lines 481-482 should be corrected (some spelling errors).
Author Response
Hello dear reviewer.
I hope you are well in this CoViD pandemic situation.
Thank you for taking the time to read and evaluate our article entitled: Total aseptization of semen, to increase the biosecurity of reproduction in swine.
All your comments and suggestions are correct, we accept them and by correcting and completing them, it will increase the quality of the article which I hope you will be ready to publish.
Ref:
Although the paper is interesting, it needs an extensive revision prior its publication in Molecules:
Answer:
We appreciate that you found our manuscript interesting, and know that it was difficult to accomplish due to the multitude of steps required for each experiment, a lot of time allocated and access to specialized pig farms.
All changes requested by you have been processed and we modified all suggested items.
Ref:
Abstract
It is not properly clear which animal they used for they study (pig and boar are both indicated); I suggest to add the scientific name of the species;
Answer:
Our study was done on pig farms that raise sows and boars. I used to collect sperm domestic boar (Suus scrofa domesticus). We have completed it with the scientific name.
In line 12, it should be better to indicate the exact number of the collected samples;
Answer:
Done, one hundred and one semples
In line 13, “determined by determining” should be changed;
Answer:
Done, determined by evaluating
In line 16, the exact number of bacterial and fungal species should be given.
Answer:
24 germs (15 bacterial and 9 fungal species) were isolated
Ref:
Introduction
The cited literature in brackets should start from 1;
Answer:
The references in the text have been changed, starting with number 1 to 45.
In line 29, please specify the meaning of “in term of metabolism”;
Answer:
We know that bacteria and fungi consume glucose as for survival and multiplication, sperm need glucose from the extender to provide the energy resources essential for mobility. The more such germs there are in the doses of AI, the more trophic competition is created that can become toxic to sperm. In the text: in terms of metabolism - I mean the relationship between germs - glucose - sperm.
In line 32, add more details concerning these “direct and indirect” deleterious effects on sperm;
Answer:
Direct damage to sperm occurs by mechanical adhesion of colonies of bacteria or hyphae that lead to agglutination of sperm in doses, and the indirect effect refers to metabolic secretions secreted by germs, which can become toxic and kill the seminal cell.
In the last part, the aim of the study should be better explained
Answer:
Done. This study aimed to highlight the bacterial and fungal contamination of boar semen during collection and the possibilities of complete aseptization of AI doses in sows, without negatively altering fertility.
Ref:
Results
In Table 1, it should be indicated if the data refers to extender A or B (and, in case, if there were differences between them),
Answer:
Done. It was mentioned in the table that in farm 1 extender A was used, and in farm 2 extender B. The difference between the extenders is: A- for long term and contains several antibiotics (the manufacturer did not supply the ingredients), and B- for permen medium and contains only one antibiotic.
In Table 2, it should be interesting if the authors could add also a quantitative data (how many coli etc);
Answer:
Done. I entered a column in the symbolizing table number of identifications (n).
The Table 4 is confusing, since it is not clear the choice of the used criteria;
Answer:
Table 4 has been deleted
In Table 7, it should be easier for the reader if, instead of T1, T2 and T3, the exact time would be indicated;
Answer:
Updated. The acronyms T1-3 have been replaced with the exact times.
In the results, the statistical analysis is completely missing (but the authors cite it in line 182);
Answer:
Statistical analyses were performed, their result was included in the discussion results and in the material and method section.
Ref:
Discussion
Lines 200-206 are results, so they should be put elsewhere,
Answer:
They were relocated at the beginning of the results section
moreover, lines 293-300 are repetition of the results and they can be avoided in this section;
Answer:
Done. Relocated in the result section, because the results are presented by percentage correlations between the experimental groups and the control group.
In line 289, the year of the cited study should be added;
Answer:
Done. was written the year 2016.
The authors should better explain why, in their opinion, 25 mg of Fluconazole induce a decrease in sperm motility (line 302);
Answer:
Different concentrations of Fluconazole were tested by probing, indicated by antifungal results. At a higher concentration (25 mg%) all sperm mobility determinations had the lowest values for both the two extenders, but also in the conservation dynamics. Because the lower concentrations were close to the control, we consider that a concentration around 25 mg% leads to the biochemical change of the sperm environment, which ultimately reduces their total mobility.
Since the authors did not find difference in sperm quality or reproductive evidence (line 317), they should provide more explanation on the aim and significance of their study;
Answer:
Comments included at the end of this section of the manuscript were provided.
The purpose of the study was to find a way to reduce and eliminate contamination of boar semen. Through the developed hygiene protocol and the addition of diluents, we managed to control the biosecurity of a boar semen. By the fact that the reproduction indices of the sows (fecundity and prolificity) were similar, the total asepticization is validated and we obtain a safe semen, destined for the conservation and sowing of the sows.
Ref:
Materials and Methods
In line 338, please explain the meaning of S-V (I suppose it should be S-W, south-west);
Answer:
Yes, it was a typo. Corrected with W
In line 371, the meaning of “when needed” should be clarified;
Answer:
Because the CASA determinations are not really accessible to everyone, we on farms, to all the boars collected, we made these determinations randomly and when the results of the classical spermogram determinations indicate differences. This is what the term "need" refers to. Howevwer the term was erased and reformulated.
The sentence in line 432 seems truncated;
Answer:
The sentence has been completed.
It is not clear why the authors did not use all the samples for all the experiments (i.e. they used just the samples from farm 1for Fluconazole experiment, and just animals from farm 3 for AI).
Answer:
Our study was conducted over several years and was a large consumer of resources, sometimes we also faced accessibility to farms and biological material due to strict biosecurity in the last period of time. The experiments were cascading and I wanted to apply them in similar farms and without disturbing the basic economic flow. Currently all 3 farmers enjoy our results as a reward for research, especially since they were not funded from any research project.
Ref:
Conclusion
The sentence in lines 481-482 should be corrected (some spelling errors).
Done.
We appreciate and we would like to thank you for your useful suggestions.
Best regards from Romania (from the Carpathian-Danubian-Pontic space)
Ștefan Gregore CIORNEI
Reviewer 2 Report
I thoroughly enjoyed reading this article and the topic is of interest, relevant to the journal and well presented. I have made some suggestions in the text of the attached file.
However, there is no statistical assessment of any of the data presented, and so I sadly must return a rejection until this is rectified and the discussion re-written to reflect the analysed results.

Author Response
Hello dear reviewer,
I hope you are well in this CoViD pandemic situation.
Thank you for spent time to read and evaluate our article entitled:
Total aseptization of semen, to increase the biosecurity of reproduction in swine.
Thank you very much for liking my article and for considering the topic of interest. It was difficult to complete this work with a great need for resources, in the situations of the two pandemics (both in pigs and in humans).
All your suggestions we accepted and corrected were completed in the text. Statistical assessment was also performed, the results improved and commented.
We appreciate and we would like to thank you for your useful suggestions.
Regards from Romania (from the Carpathian-Danubian-Pontic space)
Ștefan Gregore CIORNEI
Round 2
Reviewer 1 Report
The authors responded to all the raised issues, greatly improving the quality of the MS. It is suitable for publication in Molecules in its current form.
Author Response
Thank you for your promptness and time spent publishing this article
Reviewer 2 Report
This article is now improved but still requires major revision in the statistical analysis and descriptions. Also other changes are noted in the document. The reference list requires editing to be consistent and correct.

Author Response
Dear reviewer,
Thank you for spending time to evaluate our article entitled: Total aseptization of semen, to increase the biosecurity of reproduction in swine.
All your comments and suggestions are welcome and will improve the quality of our manuscript. We made in the text the corrections you recommended.
